# Effects of Chrysoeriol on Adipogenesis and Lipolysis in 3T3-L1 Adipocytes

**DOI:** 10.3390/foods12010172

**Published:** 2022-12-30

**Authors:** Jinhee Song, Hana Lee, Huijin Heo, Junsoo Lee, Younghwa Kim

**Affiliations:** 1Department of Food Science and Biotechnology, Chungbuk National University, Cheongju 28644, Republic of Korea; 2Department of Food Science and Biotechnology, Kyungsung University, Busan 48434, Republic of Korea

**Keywords:** chrysoeriol, adipogenesis, lipolysis, AMPK pathway, adipocytes

## Abstract

We examined the effect of chrysoeriol on adipogenesis and lipolysis and elucidated the underlying molecular mechanisms. Chrysoeriol inhibited fat deposition in adipocytes. Treatment with chrysoeriol suppressed the expression of peroxisome proliferator-activated receptor γ, fatty acid synthase, fatty acid-binding protein, CCAAT/enhancer-binding proteins (C/EBP) α, C/EBPβ, and sterol regulatory element-binding protein-1. In addition, chrysoeriol significantly elevated the activation of 5′-adenosine monophosphate-activated protein kinase. Moreover, chrysoeriol increased free glycerol and fatty acid levels and promoted lipolysis in adipocytes. Overexpression of adipose triglyceride lipase and hormone-sensitive lipase by chrysoeriol led to increased lipolysis in 3T3-L1 adipocytes. Taken together, chrysoeriol showed anti-adipogenic and lipolytic properties in adipocytes.

## 1. Introduction

Obesity is a type of metabolic disorder defined as excessive accumulation of fat in adipocytes due to overconsumption of food and lack of energy expenditure [1]. Obesity is related with various health problem, including atherosclerosis, Type 2 diabetes, hypertension, cancers and several immune-mediated disorders, such as asthma [2]. Excessive energy intake increases the quantity of lipids in the adipocytes their number—also known as adipogenesis [3,4]. Adipocytes are modulated by various transcription factors, including peroxisome proliferator-activated receptors (PPARs) and CCAAT/enhancer-binding protein (C/EBP) family [5]. These transcription factors such as C/EBPs and PPARγ are well-known for coordinating the expression of genes involved in lipid accumulation, lipolysis and fatty acid transport including adipocyte fatty acid-binding protein (aP2), adipose triglyceride lipase (ATGL), lipoprotein lipase (LPL), carnitine palmitoyltransferase-1b (CPT-1), fatty acid synthase (FASN), leptin, and hormone-sensitive lipase (HSL) [6,7]. Lipolysis is a catabolic process that leads to the liberation of fatty acid (FA) from triacylglycerol (TG) [8]. In lipolysis, TG is hydrolyzed to glycerol and FAs in a series of successive reactions involving ATGL, which converts TG into diglyceride (DG), HSL, which converts DG into monoglyceride (MG) and MG lipase, which converts MG into FAs and glycerol [9]. Impaired lipolysis in adipocytes may be related to clinical symptoms including insulin resistance, diabetes mellitus, obesity, and dyslipidemia [10]. Therefore, lipolysis would be an important therapeutic target of obesity management and treatment.

Recently, it has been reported that the 5′-adenosine monophosphate-activated protein kinase (AMPK) signaling pathway is well-known as a critical role in modulating energy balance. Acetyl-CoA carboxylase (ACC) is a critical enzyme that is associated with fatty acid oxidation and biosynthesis [11]. Upregulation of the AMPK pathway inhibits the expression of C/EBPα, sterol regulatory element-binding protein-1 (SREBP-1), and PPARγ and, consequently, suppresses fat deposition during adipogenesis [12]. Therefore, targeting the AMPK pathway has been proposed as a novel strategy to prevent metabolic disorders.

Rooibos tea (*Aspalathus linearis*) is well-known for treating hypertension, asthma, and diarrhea [13]. Chrysoeriol is a major flavonoid in Rooibos tea [13]. Chrysoeriol is one of the flavonoids found in several tropical medicinal plants [14]. Among the numerous functions and bioactivities it possesses are its anti-obesity, anti-inflammatory, and anti-mutagenic properties [15,16,17,18]. Chrysoeriol directly scavenges superoxide radicals from xanthine oxidase system [19]. Moreover, chrysoeriol has been shown to control the immune system, reduce obesity, and relax smooth muscles [20]. Although diverse bioactivities of chrysoeriol have been reported, the anti-obesity mechanisms of chrysoeriol remain unclear. Therefore, the objectives of present study were to elucidate the effects of chrysoeriol on adipogenesis and lipolysis in adipocytes.

## 2. Materials and Methods

### 2.1. Chemicals

Chrysoeriol was purchased from Toronto Research Chemicals (Toronto, ON, Canada). Antibodies against C/EBPα, C/EBPβ, PPARγ, SREBP-1c, FAS, β-actin, and aP2 were obtained from Santa Cruz Biotechnology (Santa Cruz, CA, USA) and those against p-ACC, ACC, p-AMPK, AMPK, p-HSL(Ser563), HSL, and ATGL were purchased from Cell Signaling Technology (Danvers, MA, USA). Dimethyl sulfoxide (DMSO), insulin, dexamethasone, and isobutyl methyl xanthine were purchased from Sigma Chemicals Co. (St. Louis, MO, USA).

### 2.2. Cell Culture and Adipocytes Differentiation

The 3T3-L1 cells were obtained from the American Type Culture Collection (Manassas, VA, USA). The 3T3-L1 cells were differentiated as described in a previous report [12]. The experimental design of the present study is shown in Figure 1. 

### 2.3. Cytotoxicity and Fat Deposition 

An MTT assay was conducted to evaluate the cytotoxicity of chrysoeriol in adipocytes. 3T3-L1 preadipocytes were cultured without (control group) or with chrysoeriol (6.25, 12.5, and 25 μM) in the differentiation medium (MDI, 1 μM dexamethasone + 0.5 mM 3-isobutyl-1-methylxanthine + 1 μg/mL insulin) for 6 days, and MTT reagent was added into each well. After 2 h, the culture medium in the well was discarded, and the blue-crystalized formazans were dissolved in DMSO. Formazan solution was quantified at 550 nm using a microplate reader (BioTek, Inc., Winooski, VT, USA). Fat deposition was measured using Oil Red O staining [21].

### 2.4. Measurement of Free Glycerol and Fatty Acid

Free glycerol content was measured by an adipolysis assay kit (Cayman Chemical, Ann Arbor, MI, USA) following the manufacturer’s protocol. Free fatty acid levels were assayed using a colorimetric kit (Biovision, Milpitas, CA, USA). 

### 2.5. Western Blot Analysis

Cell lysates were obtained using a Pro-PrepTM sample buffer (iNtRON Biotechnology, Seongnam, Republic of Korea). Proteins (50 μg/lane) were separated on SDS-PAGE gel and transferred to nitrocellulose membrane. Skim milk solution (5%) was used for blocking, and membranes and membranes were incubated with diluted antibodies at 4 °C overnight. Protein bands were detected using the SuperSignal™ West Pico (Thermo Fisher Scientific, Waltham, MA USA), and protein bands were analyzed using ImageJ Software version 1.53t (NIH, Bethesda, MD, USA).

### 2.6. Statistical Analysis

GraphPad Prism 5 software (GraphPad Software, San Diego, CA, USA) was used to calculate the statistics, and all data were analyzed using a one-way analysis of variance (ANOVA) followed by Tukey’s test. Data are presented as the mean ± standard error. Values are expressed as the mean ± SE (n = 3).

## 3. Results and Discussion

### 3.1. Effects of Chrysoeriol on Cytotoxicity and Fat Accumulation in Adipocytes

The cytotoxicity of chrysoeriol was assessed using the MTT assay. Chrysoeriol was not cytotoxic at concentrations up to 25 μM in the 3T3-L1 cells for 6 days (Figure 2a). Lipid droplets were evaluated using Oil Red O staining. In the results, treatment with chrysoeriol reduced fat accumulation in adipocytes compared with that in the control (Figure 2b). Lipid accumulation was measured by optical density values after Oil Red O staining. Chrysoeriol reduced lipid deposition in adipocytes in a concentration-dependent manner. Increased lipid accumulation inside adipocytes is a typical form of metabolic failure in obesity [4]. Therefore, dietary interventions using phytochemicals have been used to inhibit lipid accumulation in adipocytes and improve obesity. Several flavonoids, such as quercetin and kaempferol, are known to reduce intracellular lipid levels in adipocytes [22,23]. These results showed that chrysoeriol decreased fat deposition in adipocytes and could inhibit adipogenesis.

### 3.2. Effect of Chrysoeriol on the Expressions of Adipogenic Markers

To evaluate the effect of chrysoeriol on the expression of adipogenic markers such as PPARγ, C/EBPs and SREBP1c, chrysoeriol-treated 3T3-L1 adipocytes were collected on day 6. As shown in Figure 3, chrysoeriol markedly inhibited the protein expression of PPARγ, C/EBPα, C/EBPβ, and SREBP1. Generally, adipogenesis is accompanied by changes in gene expression, cell morphology, and hormone sensitivity [24]. Adipogenesis is modulated through the combination of transcription factors and their targets. The expression of C/EBPα, C/EBPβ, and PPARγ is related to adipocyte differentiation [25]. SREBP1c is associated with the expression of lipogenic genes related in fatty acid synthesis [26].

Then, we investigated whether chrysoeriol inhibits the expression of late adipocyte markers such as FAS and aP2 in adipocytes. The results showed that chrysoeriol significantly suppressed aP2 and FAS protein expression compared to that in control cells (Figure 4). The genes encoding aP2 and FAS are late differentiation markers of adipocytes and play a pivotal role in the signaling pathway related to obesity. aP2 plays a key role in obesity associated with fatty acid metabolism, and the inhibition of aP2 expression is linked to the decreased utilization of fatty acids [27]. FAS, a lipogenic enzyme, regulates triglyceride synthesis and storage of triglycerides [28]. Meanwhile, it is well-known that chrysoeriol is the main methylation metabolite of luteolin in vivo [29]. In a previous study, chrysoeriol, the methylated luteolin, was more permeable in vivo due to its relatively higher hydrophobicity than luteolin, so that it might be more easily distributed into tissues [29]. Therefore, it seems that chrysoeriol could show directly anti-adipogenic effects in adipocytes, although this requires thorough investigations to elucidate the actual metabolism of chrysoeriol. These results showed that chrysoeriol decreases lipid deposition via the modulation of adipogenic markers and their downstream targets.

### 3.3. Effect of Chrysoeriol on AMPK Signaling Pathway

To elucidate the mechanism underlying the anti-adipogenenic effect exerted by chrysoeriol through the AMPK pathway, the phosphorylated AMPK was measured. Chrysoeriol activated AMPK in a concentration-dependent manner (Figure 5a). At the highest chrysoeriol concentration (25 μM), the level of phosphorylated AMPK/total AMPK was approximately two-fold higher than that in the control adipocytes. Acetyl CoA carboxylase (ACC) is one of the major metabolic targets of AMPK; hence, the effect of chrysoeriol on ACC was elucidated by measuring ACC deactivation and phosphorylated ACC levels. The results showed that the levels of phosphorylated ACC were significantly increased approximately 2.5-fold by treatment with chrysoeriol (25 μM) (Figure 5b). AMPK is a critical target for the management of obesity because it is involved in glucose uptake, insulin sensitivity, and fatty acid oxidation in adipocytes. Moreover, AMPK is a sensor of energy balance, and its activation decreases lipid synthesis via the suppression of ACC [30]. An earlier study reported that inactive phosphorylated ACC inhibits adipogenesis [31]. Natural compounds such as ursolic acid and berberine have been recently reported to inhibit lipid accumulation via upregulation of the AMPK pathway in adipocytes [32,33]. In addition, several studies have shown that natural substances such as nobiletin and indole derivatives exhibit anti-adipogenic effects via modulation of the AMPK in adipocytes [12,34]. Chrysoeriol led to the phosphorylation of AMPK and ACC, which might reduce fat accumulation in adipocytes.

### 3.4. Effect of Chrysoeriol on Lipolysis

To evaluate whether chrysoeriol induces lipolysis, the adipocytes were treated with chrysoeriol. The results on day 7 showed that chrysoeriol significantly increased the release of glycerol and free fatty acid in adipocytes (Figure 6). Lipolysis in adipocytes is an important metabolic pathway for the degradation of TG to fatty acids and glycerol to generate energy [35]. In addition, lipolytic processes result in reduced fat content in adipose tissue [35]. Lipolysis in adipocytes is modulated by the action of ATGL and HSL [36]. 

Next, we investigated whether chrysoeriol regulates its lipolytic effects by modulating these lipases and examined the expression of ATGL and HSL in mature adipocytes. The phosphorylation of HSL at Ser563 was increased in chrysoeriol-treated adipocytes (Figure 7a). Chrysoeriol treatment for 24 h also up-regulated ATGL protein expression (Figure 7b). Previous studies have reported that ATGL and HSL are primarily responsible for TG hydrolysis in adipose tissues [37]. To complete TG hydrolysis, ATGL interacts with its coactivator congenital generalized lipodystrophy-58 protein in the cytosol, whereas HSL translocates to the lipid droplet when activated and is involved in lipolysis [38]. HSL is stimulated by a number of hormones and signaling pathways and can be activated at various sites. In a previous study, increased lipolysis was regulated through the phosphorylation of HSL at Ser563 and Ser660 and the activation of perilipin, which is a key regulator of lipolysis [39]. These results show that the increased protein expression of ATGL and HSL by chrysoeriol enhances lipolysis in adipocytes. The treatment of chrysoeriol provides evidence for the positive modulation of adipogenesis and lipolysis in adipocytes by chrysoeriol.

## 4. Conclusions

In the present study, chrysoeriol inhibited adipogenesis by suppressing the expression of transcription factors and adipogenic proteins (FAS and aP2) in 3T3-L1 cells. In addition, chrysoeriol induces the activation of AMPK and ACC during adipogenesis. Moreover, chrysoeriol stimulated the release of free fatty acids and glycerol through the upregulation of ATGL and HSL. Taken together, chrysoeriol exerts anti-adipogenic effects by regulating adipogenic transcriptional factors through phosphorylation of the AMPK pathway and lipolysis in adipocytes. Finally, this study confirms the potential health-benefit of chrysoeriol on adipocytes, but further in-vivo research (e.g., nutritional interventions) would be needed to elucidate the benefits of consuming chrysoeriol-rich foods, such as Rooibos tea, on obesity and its comorbidities.

## Figures and Tables

**Figure 1 foods-12-00172-f001:**
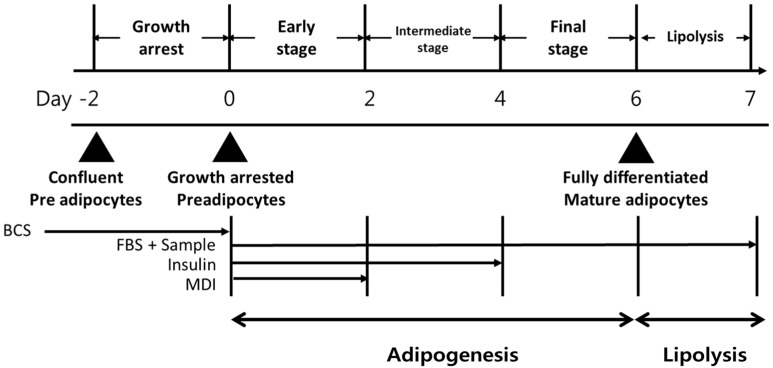
Scheme of 3T3-L1 differentiation. 3T3-L1 preadipocytes were differentiated into adipocytes without (control group) or with chrysoeriol. BCS, bovine calf serum; FBS, fetal bovine serum; MDI, 1 μM dexamethasone + 0.5 mM 3-isobutyl-1-methylxanthine + 1 μg/mL insulin.

**Figure 2 foods-12-00172-f002:**
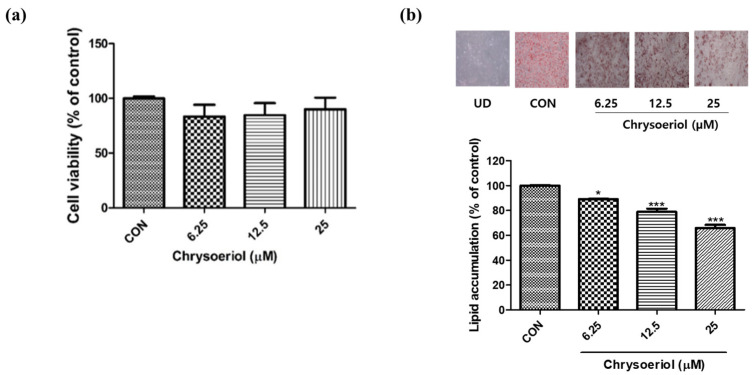
Effect of chrysoeriol on cytotoxicity (**a**) and lipid accumulation (**b**) observed using Oil-red O staining in 3T3-L1 adipocytes on day 6. *, *** *p* < 0.05 and 0.001, respectively, significant difference compared with control group. UD, undifferentiated cells; CON, control (differentiated cells).

**Figure 3 foods-12-00172-f003:**
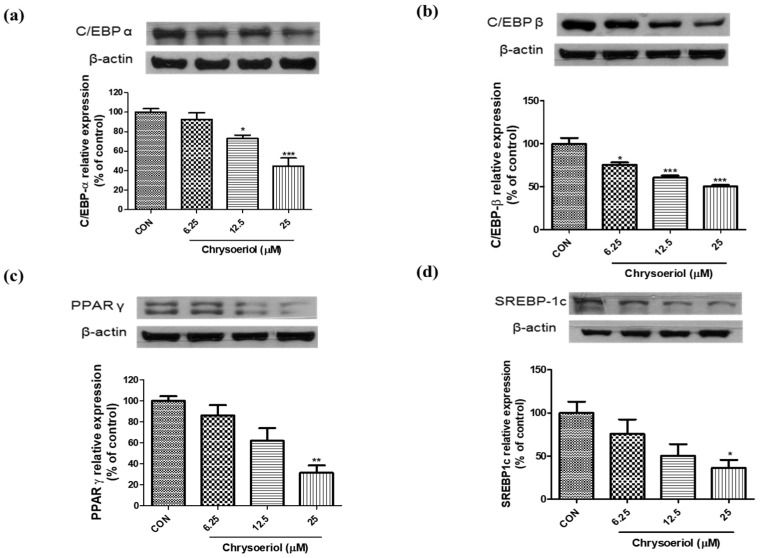
Effects of chrysoeriol on CEBP-α (**a**), C/EBP-β (**b**), SREBP1c (**c**) and PPARγ (**d**) expression in 3T3-L1 adipocytes. *, **, *** *p* < 0.05, 0.01, and 0.001, respectively, significant difference compared with control group. CON, control (differentiated cells).

**Figure 4 foods-12-00172-f004:**
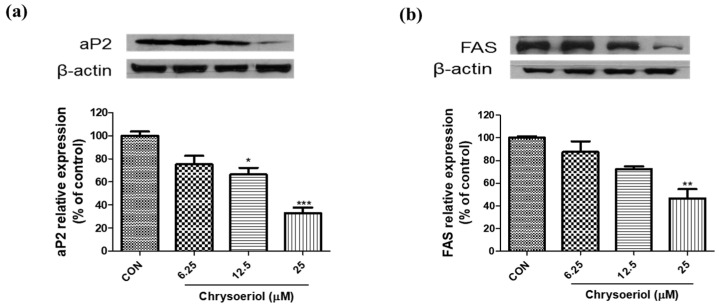
Effects of chrysoeriol on aP2 (**a**) and FAS (**b**) expression in 3T3-L1 adipocytes. *, **, *** *p* < 0.05, 0.01, and 0.001, respectively, significant difference compared with control group. CON, control (differentiated cells).

**Figure 5 foods-12-00172-f005:**
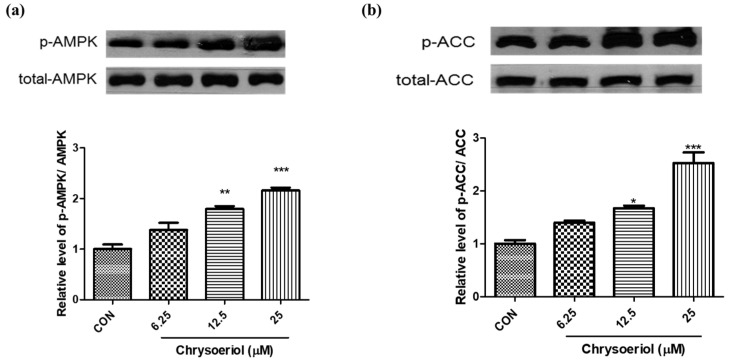
Effects of chrysoeriol on monophosphate-activated protein kinase (AMPK) (**a**) and acetyl-CoA carboxylase (ACC) (**b**) phosphorylation in 3T3-L1 adipocytes. *, **, *** *p* < 0.05, 0.01, and 0.001, respectively, significant difference compared with control group. CON, control (differentiated cells).

**Figure 6 foods-12-00172-f006:**
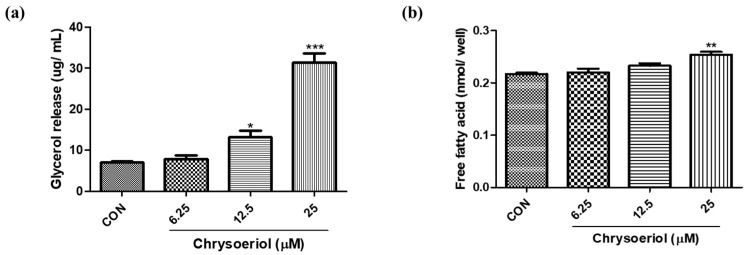
Effect of chrysoeriol on release of free glycerol (**a**) and free fatty acids (**b**) in 3T3-L1 adipocytes on day 7. *, **, *** *p* < 0.05, 0.01, and 0.001, respectively, significant difference compared with control group. CON, control (differentiated cells).

**Figure 7 foods-12-00172-f007:**
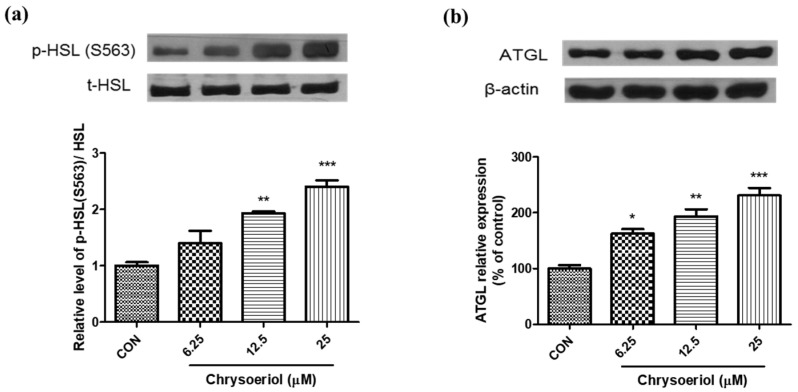
Effects of chrysoeriol on HSL phosphorylation (**a**) and ATGL expression (**b**) in 3T3-L1 adipocytes. *, **, *** *p* < 0.05, 0.01, and 0.001, respectively, significant difference compared with control group. CON, control (differentiated cells).

## Data Availability

The data that support the findings of this study are available from the corresponding author upon reasonable request.

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
