# Peer review of "Effects of Chrysoeriol on Adipogenesis and Lipolysis in 3T3-L1 Adipocytes"

_foods, 2022, doi:10.3390/foods12010172_

Round 1

Reviewer 1 Report

The researchers investigated chrysoeriol effect on adipogenesis and lipolysis by assessing in 3T3-L1 2 Adipocytes. Hypothesis of an anti-obesity effect of chrysoeriol was questioned. The results reveal the potential of a natural supplement to combat people's overweight and obesity  since chrysoeriol is found in medicinal plants.

Adequate methods have been chosen for the research. 

Line 77:   Description of the control sample is recommended to add. 

The references seem to be appropriate. Please check the references and add DOI, where it is missing (ex., line 251, etc.). Please delete double numbers 13 and 14 (references).

In the Conclusion, the effect of chrysoeriol on adipogenesis and lipolysis is described. And in the Introduction, the Authors discuss the presence of chrysoeriol in tropical medicinal plants, such as Rooibos tea (Aspalathus linearis), but not in the Conclusion. It would be interesting to learn the Author's opinion on how would the obtained data be used for obesity treatment with natural ingredients.

Author Response

Thank you very much for your consideration, and we really appreciate the comments and have learned a lot. We have revised the manuscript according to your suggestions. The changes were marked with red color in the revised manuscript. And our answers are noted below.

1. Line 77:   Description of the control sample is recommended to add. 

A: We added the description of the control in line 76-77.

“3T3-L1 preadipocytes were differentiated into adipocytes without (control group) or with chrysoeriol.”

2. The references seem to be appropriate. Please check the references and add DOI, where it is missing (ex., line 251, etc.). Please delete double numbers 13 and 14 (references).

A: We added DOI except for Ref. #2 (not available), and delete the reference 14.

3. In the Conclusion, the effect of chrysoeriol on adipogenesis and lipolysis is described. And in the Introduction, the Authors discuss the presence of chrysoeriol in tropical medicinal plants, such as Rooibos tea (Aspalathus linearis), but not in the Conclusion. It would be interesting to learn the Author's opinion on how would the obtained data be used for obesity treatment with natural ingredients.

A: Rooibos contains various phytochemicals (such as aspalathin, luteolin, and quercetin) which have anti-obesity effects in addition to chrysoeriol. Also, Rooibos tea has displayed anti-hypertensive, anti-inflammatory, antioxidant, anti-diabetic, anti-viral, cardiometabolic and anti-obesity effects in previous studies. Therefore, we added the below concluding sentence in conclusion part.

“Finally, this study confirms the health beneficial effect of chrysoeriol and the consumption of chrysoeriol rich food such as Rooibos tea would be expected to have favorable effects on obesity.”

Reviewer 2 Report

In this paper, the authors investigated effects of chrysoeriol on adipogenesis and lipolysis in 3T3-L1 adipocytes. This is a very interesting research. What the results indicate is clear and the description is almost fine. Some questions are list below.

 1. Line 50, “Rooibos tea (Aspalathus linearis) is well-known for …” instead of “Rooibos tea is a well-known for …”.

 2. Line 51, “rooibos tea” instead of “Rooibos tea (Aspalathus linearis)”.

 3. Line 126, “expressions” instead of “expression”.

 4. Line 199, “provides” or “provided” instead of “provide”.

 5. The authors investigated the effects of chrysoeriol on adipocytes. Can chrysoeriol directly touch the adipocytes of organism, or in the form of metabolite? In my opinion, the effects of metabolite of chrysoeriol on adipocytes should be considered.

 6. Figure 7b, the Y-axis indicates “ATGL relative expression (% of control)”. Why the relative expression of CON group is not 100%? Please explain.

Author Response

Thank you very much for your consideration, and we really appreciate the comments and have learned a lot. We have revised the manuscript according to your suggestions. The changes were marked with red color in the revised manuscript. And our answers are noted below.

1. Line 50, “Rooibos tea (Aspalathus linearis) is well-known for …” instead of “Rooibos tea is a well-known for …”.

A: It was revised according to your comment.

2. Line 51, “rooibos tea” instead of “Rooibos tea (Aspalathus linearis)”.

A: It was revised according to your comment.

3. Line 126, “expressions” instead of “expression”.

A: It was revised according to your comment.

4. Line 199, “provides” or “provided” instead of “provide”.

A: It was revised according to your comment.

5. The authors investigated the effects of chrysoeriol on adipocytes. Can chrysoeriol directly touch the adipocytes of organism, or in the form of metabolite? In my opinion, the effects of metabolite of chrysoeriol on adipocytes should be considered.

A: We added the below sentence in discussion part (line 149-155).

It is well-known that chrysoeriol is the main methylation metabolite of luteolin in vivo. In previous study, chrysoeriol, the methylated luteolin, was more permeable in vivo due to their relatively higher hydrophobicity than luteolin, so that might be more easily distributed into tissues. Therefore, it seems that chrysoeriol could show directly the anti-adipogenic effects in adipocytes, although this requires thorough investigations to elucidate the actual metabolism of chrysoeriol at play.

6. Figure 7b, the Y-axis indicates “ATGL relative expression (% of control)”. Why the relative expression of CON group is not 100%? Please explain.

A: It was our mistake. We revised that. Thank you for your comment.
